# Effect of Immune Stress on Growth Performance and Immune Functions of Livestock: Mechanisms and Prevention

**DOI:** 10.3390/ani12070909

**Published:** 2022-04-02

**Authors:** Xueting Niu, Yuexia Ding, Shengwei Chen, Ravi Gooneratne, Xianghong Ju

**Affiliations:** 1Department of Veterinary Medicine, Guangdong Ocean University, Zhanjiang 524088, China; nxt1208@163.com (X.N.); dingyuexia2006@163.com (Y.D.); csw9610@163.com (S.C.); 2Marine Medical Research and Development Centre, Shenzhen Institute of Guangdong Ocean University, Shenzhen 518018, China; 3Faculty of Agriculture and Life Sciences, Lincoln University, Lincoln 7647, New Zealand; ravi.gooneratne@lincoln.ac.nz

**Keywords:** immune stress, immune function, growth performance, mechanisms, prevention

## Abstract

**Simple Summary:**

Immune stress is an important stressor in domestic animals that leads to decreased feed intake, slow growth, and reduced disease resistance of pigs and poultry. Especially in high-density animal feeding conditions, the risk factor of immune stress is extremely high, as they are easily harmed by pathogens, and frequent vaccinations are required to enhance the immunity function of the animals. This review mainly describes the causes, mechanisms of immune stress and its prevention and treatment measures. This provides a theoretical basis for further research and development of safe and efficient prevention and control measures for immune stress in animals.

**Abstract:**

Immune stress markedly affects the immune function and growth performance of livestock, including poultry, resulting in financial loss to farmers. It can lead to decreased feed intake, reduced growth, and intestinal disorders. Studies have shown that pathogen-induced immune stress is mostly related to TLR4-related inflammatory signal pathway activation, excessive inflammatory cytokine release, oxidative stress, hormonal disorders, cell apoptosis, and intestinal microbial disorders. This paper reviews the occurrence of immune stress in livestock, its impact on immune function and growth performance, and strategies for immune stress prevention.

## 1. Introduction

Stress refers to a series of nonspecific responses when the body is stimulated by external and internal abnormal stimuli [1]. In animal husbandry, common stresses include housing density, weaning, immunization, capture, transportation, and exposure to ammonia, temperature, humidity, noise, and high doses of pathogens [2]. Different stresses cause specific biological changes and endanger the animal’s health. Immune stress (IS) is also called immune stimulation. Specifically, in immunology, it refers stimulation of the body by an antigen eliciting an immune response. In a narrow sense, it refers to the systemic nonspecific adaptive immune response following inoculation with an antigen. In a broad sense, it refers to the frequent infection of various pathogenic microorganisms in the body under poor sanitary conditions, resulting in continuous activation of the immune system by invading pathogenic microorganisms and eliciting a nonspecific adaptive response [3]. 

In recent years, due to the large-scale and high-density farming environment and livestock genetic improvement, animals are vulnerable to a range of pathogens. Therefore, animals require to be vaccinated more frequently. Immunization can protect livestock from pathogens, but it can also stimulate immunity as a special stress factor, which can lead to a series of adverse phenomena such as fever, depression, anorexia, intestinal microflora disorders, nutritional and metabolic changes, and reduced growth, culminating even in increased mortality, resulting in a huge economic loss to farmers and the industry in general. This paper reviews the effects and mechanisms of IS on immune function and growth performance of livestock and poultry and provides a basis for revealing the molecular mechanism of IS. 

## 2. Mechanism of IS on the Neuroendocrine–Immune System

Immune system activation leads to the release of cytokines (interferon-γ (IFN-γ), interleukin-6 (IL-6), interleukin-1β (IL-1β), nitric oxide (NO), prostaglandin E2 (PGE2), tumor necrosis factor (TNF-α), etc.) by monocytes, macrophages, and lymphocytes [4,5,6,7]. The classical lipopolysaccharide (LPS)-induced IS model has shown that the release of inflammatory cytokines is closely related to toll-like receptors (TLRs) found on the immune cells’ surface [8,9,10]. When stimulated by an immunogen, the TIR region of TLR4 binds to the carboxyl end of myeloid differentiation factor 88 (MyD88), and its amino end binds to the IL-1 receptor-associated kinase (IRAK). After phosphorylation, IRAK is separated from MyD88 and released into the cytoplasm. Activated IRAK then binds to TNF receptor-related factor 6 (TRAF-6) and connects with nuclear factor κB (NF-κB)-inducible kinase (NIK), further activating the IκB kinase complexes (IKKs). Activated IKKs act on IκB t, ubiquitinating and degrading, thus activating the NF-κB. Meanwhile, the mitogen-activated protein kinase (MAPK) signaling pathway is stimulated, resulting in activation of cytoplasmic protein 1 (AP-1) which then enters the nucleus to induce the expression of a variety of inflammatory cytokines and chemokines, macrophage migration, and phagocytosis [11,12,13,14,15,16]. In addition to TLR4-NF/κB and MAPK/AP-1 signaling pathways, the JAK/STAS signaling pathway is also involved in the release of proinflammatory factors during IS [17].

Cytokines released by peripheral immune cells subsequently enter the central nervous system (CNS) to activate local immunity through several mechanisms, including (1) the active uptake mechanism via the blood–brain barrier [18], (2) stimulation of the leakage region in the blood–brain barrier of circumventricular organs [19], (3) influencing the afferent neurons of the peripheral vagus nerve to transmit cytokine signals to nerve cells in relevant brain regions, including nucleus tractus solitarius and hypothalamus (also known as the “neural pathway”) [20,21,22], (4) activation of the endothelial cells and microglia in the cerebrovascular system to produce local inflammatory mediators such as inflammatory cytokines, chemokines, cyclooxygenase-2(COX-2), prostaglandin E2 (PGE2), and NO [23,24], (5) activation of immune cells such as monocytes/macrophages and T cells recruited from the periphery to the brain parenchyma via monocyte chemoattractant protein-1 (MCP-1) to produce more cytokines and inflammatory mediators in the brain [25,26]. After entering the CNS, these peripheral cytokine signals are amplified by the local inflammatory network (including the inflammatory signal transduction pathway, cytokines, COX-2 expression, and PGE2 release), so that the CNS produces a large number of inflammatory cytokines (such as TNF-α, IL-1 β, IL-6, IFN-γ, etc.) [23]. Furthermore, a large number of inflammatory factors induce excessive production of reactive oxygen species (ROS) which cause oxidative stress [27] and enlarge central nervous inflammation [28].

The hippocampus in the HPA axis is sensitive to stress injury. The sharp increase in inflammatory cytokines coupled with oxidative stress lead to dendritic atrophy and neuronal death in the hippocampus [29], followed by activation of the HPA resulting in the synthesis of corticotropin-releasing hormone (CRH) and arginine vasopressin (AVP) by the intermediate small cell neurons in the hypothalamic paraventricular nucleus. These secretagogues are released to the anterior pituitary through the portal blood. CRH acts on the pituitary and binds to CRHR1 to promote the synthesis and secretion of pro-opioid cortisol (POMC) in the anterior pituitary. POMC is catalyzed by the prohormone-converting enzyme, resulting in the release of adrenocorticoid hormone (ACTH) from the anterior pituitary. ACTH in the bloodstream activates the adrenal melanocortin receptor 2 (MC2R) to stimulate adrenal cortical cells to synthesize and release cortisol (COR) and other glucocorticoids (GC) [6,30].

There are abundant glucocorticoid receptors in the hippocampus which can inhibit the stress response of the HPA axis when bound to glucocorticoid [31,32,33]. In contrast, COR can suppress the immune system by targeting genes related to cytokines, chemokines, inflammatory proteins and their receptors, and also affect cell adhesion. At the transcriptional level, COR binds to its receptors, inhibits the transcriptional activity of AP-1 and NF-κB, thereby inhibiting the expression of various cytokines [34]. It can also directly act on immune cells such as macrophages, mast cells, and basophils, thereby inhibiting their immune function, inducing apoptosis of B and T lymphocytes and dendritic cells, inhibiting their proliferation, differentiation, and migration, reducing humoral and cellular immune function, and causing immunosuppression [35,36]. The mechanism of IS effects on the neuro–endocrine–immune system is shown in Figure 1. 

### 2.1. IS Leads to Immune Dysfunction of Livestock 

The effect of IS on animal immune function is bidirectional. It may enhance or suppress immune function, which depends largely on stress intensity and duration of stress [37]. It is generally believed that short-term (acute) stress can enhance the immune function of livestock, while long-term, low-intensity (chronic) stress can cause immunosuppression [37,38].

#### 2.1.1. Acute IS Causes Enhanced Immune Function of Livestock 

IS significantly increases the plasma concentrations of COR, PEG2, IL-6, TNF-α, and IL-1β [39]. IS induced by LPS increases IL-1, IL-2, IL-6, immunoglobulin G (IgG), and immunoglobulin A (IgA) levels in broiler serum [40]. LPS-induced IS can strongly inhibit C4BPA (a cycle inhibitor of the classical and MBL pathways of the complement system activation pathways) in peripheral blood mononuclear cells (PBMC) in pigs, thereby enhancing the activation of the complement system [41]. Chickens vaccinated with *Eimeria tenella* express high IgG and sIgA levels [42]. LPS and human serum albumin (HuSA) used to construct a broiler IS model found that IS significantly increases total immunoglobulin, immunoglobulin M (IgM), IgA, and IgG levels in the serum, indicating that IS enhances the broiler humoral immune response [43]. Similarly, IS significantly increases serum IgA levels in Linwu ducks [44], the spleen and thymus index of broilers [45], while LPS-induced IS increases the ratio of CD4^+^/CD8^+^ in duck spleen and thymus [46]. IS can increase the concentration of IL-1β, PGE2, and COR in piglet plasma and promote the proliferation of peripheral blood lymphocytes, an indication of enhanced cellular immune function in piglets [47,48,49], similarly to another study which also showed proliferation of lymphocytes and increased ratio of CD4^+^/CD8^+^ in peripheral blood lymphocytes of chickens, and a significant increase in serum TNF-a, IL-6, IL-1, and IgG levels [50,51]. Thus, all of the above studies confirm that IS can enhance the immune function of livestock fairly quickly so that the body can better resist a pathogen invasion.

#### 2.1.2. Chronic IS Causes Immunosuppression in Livestock 

In contrast to acute IS, chronic IS causes suppression of cellular immunity in livestock. Chronic IS increases serum ATCH, COR, and expression of Caspase-3 and Caspase-9 in immune cells, decreases the ratio of Bcl-2 to Bax, decreases the index of immune organs (bursa, thymus, and spleen) and the number of neutrophils, CD4^+^ cells, and CD8^+^ cells, and induces apoptosis of immune cells [52,53,54]. Chronic IS also affects humoral immunity, a decrease in serum IgG levels and lysozyme activity in livestock [55,56]. Chronic IS also affects intestinal immunity. Chronic IS markedly damages the intestinal structure of livestock, enhances lymphocyte apoptosis in intestinal Peyer’s nodules, reduces intestinal IgA-secreting cells (ASC) and secretory immunoglobulin A (sIgA), causing destruction of the intestinal immune barrier [54] and also the intestinal mechanical barrier by downregulating tight junction proteins and inducing intestinal epithelial cell apoptosis, and the intestinal microbial barrier by interfering with the composition of intestinal microorganisms, with all these effects culminating in serious damage to intestinal immune function [57,58,59]. Thus, chronic IS inhibits the body’s intestinal, cellular, and humoral immunities.

## 3. Mechanism of IS on Growth Performance of Livestock and Poultry

Although the impact of acute and chronic IS based on the intensity and duration of stress is different, the effects on the livestock growth performance such as the decrease in average daily feed intake (ADFI) and average daily gain (ADG) and increased feed meat ratio (FCR) are somewhat similar.

### 3.1. Effects and Mechanism of IS on Livestock Feed Intake 

IS significantly decreases the ADFI of livestock [45,47,53,60]. It is believed that this could be due to a reduced feed intake caused by secretion of specific hormones and inflammatory factors.

The brain–gut axis plays a vital role in the loss of appetite caused by IS. The hypothalamus, especially its arcuate nucleus (ARC) that is involved in energy perception and integration. In ARC, neuropeptide Y (NPY) and, agouti-related peptide (AgRP), a neuropeptide produced by the AgRP/NPY neuron, and also gamma-aminobutyric acid (GABA) can increase food intake, while POMC, cocaine- and amphetamine-regulated transcripts (CART) reduce food intake [61]. While intestinal hormones (leptin, cholecystokinin (CCK), glucagon-like peptide 1 (GLP-1), resistin, and growth hormone-releasing peptide) can initiate most of the signal transduction and communication in the gut–brain axis, they can also regulate appetite by activating or inhibiting hypothalamic neurons [62]. 

In IS, a large number of inflammatory factors released by the peripheral immune system enter the circulation. These can induce adipocytes to synthesize and secrete leptin [63,64] and transport it to the brain. With assistance from IL-1β, leptin binds to the leptin receptor in hypothalamic neurons [65], stimulates POMC neurons in ARC neurons to produce anorexic peptide POMC, and inhibits AgRP/NPY neurons from producing appetite peptides NPY and AgRP [20,66,67]. LPS-induced IS can significantly increase resistin and GLP-1 and decrease ghrelin in animals [68,69,70]. Excess GLP-1 can overactivate GLP-1 receptors in the CNS, gastrointestinal tract (GIT), and pancreas, activate GLP-1 receptors in the CNS, all of which can lead to a fullness feeling in the animal. GLP1 can also reduce the speed of gastric emptying and hinder gastric acid secretion, thereby increasing gastric dilatation, limiting excessive food consumption, and enhancing satiety [68]. 

Ghrelin, the only known peripherally derived appetite hormone, activates the hypothalamic growth hormone secretagogue receptor (GHSR-1a) [61], mediates the activation of AgRP/NPY neurons in ARC [71], and inhibits the satiety POMC neurons [72], promoting appetite. Therefore, a decline in the level of ghrelin leads to a decrease in appetite. The mechanism of appetite suppression by IS is shown in Figure 2.

### 3.2. Effect of IS on the Digestion and Absorption of Livestock and Poultry

Mucosal integrity, digestive enzymes, transport proteins, and microbes of the intestines play a major role in the digestion and absorption of nutrients. Nutrient absorption mostly occurs in the small intestine (SI). Iintestinal villi height (VH) growth will increase the contact area between the nutrients and SI, resulting in improved absorption rate of nutrients. This is supported by shallower crypt depth (CD) which facilitates the proliferation of intestinal epithelial cells to further improve absorption from the SI. Thus, both VH and CD, and the ratio of VH/CD (known as V/C) are often used to evaluate intestinal absorptive capacity [73]. Furthermore, intestinal microbes can ferment indigestible substrates such as dietary fiber and endogenous intestinal mucus to produce short-chain fatty acids (SCFA) which are easily absorbed [74]. Thus, intestinal microbial composition is important for optimal intestinal function.

IS can seriously affect intestinal health. LPS-induced IS can reduce VH and enhance CD and reduce alkaline phosphatase (AKP) activity in the duodenum and jejunum, and also decrease the number of duodenal sodium-dependent glucose cotransporter 1 (SGLT1) and the plasma D-xylose level [75]. IS decreases the level of lactase, maltase, and sucrase markedly in the jejunum and ileum of piglets [76,77]. Furthermore, IS increases SI interleukin-8 (IL-8) secretion, ROS production, and TNF-α mRNA abundance, but decreases the level of SGLT1, excitatory amino acid transporter 1 (EAAC1), H^+^/peptide cotransport 1 (PEPT-1), and intestinal fatty acid binding protein 2 (I-FABP2) in the intestine of weaned piglets [78]. It was also found that the abundance and distribution of tight junction proteins (zonula occluden-1 (ZO-1), Occludin, and claudin-1) mRNA in the SI declines when exposed to IS [79]. IS makes duck SI walls thinner, lowers the V/C value, reduces the expression of tight junction proteins (ZO-1, Occludin, and Claudin-1), and enhances intestinal permeability [44]. IS not only downregulates the expression of digestive enzymes (maltase, AKP, and Na-K-ATPase) and tight junction proteins (CLDN-1, OCLD, ZO-1, ZO-3, EpCAM, and JAM2), but also reduces the phosphorylation of mTOR protein in the jejunum and inhibits the proliferation of jejunum cells [80]. IS causes rumen and intestinal flora disorders in livestock, resulting in SI damage, reduced intestinal flora diversity, and an increased harmful-to-beneficial bacteria ratio [56,81,82,83,84,85,86].

Intestinal injury cause inflammation, oxidation, apoptosis, and autophagy. IS induces the expression of key genes related to TLR4 and the nucleotide-binding oligodomain proteins (NODs) signaling pathway and increases the expression of inflammatory factors (IL-1β, IL-2, IL-6, IL-8, INF-γ, and TNF-α), iNOS, and COX-2 in the SI [44,76,77,78,87]. IS also reduces SI superoxide dismutase (SOD) and glutathione peroxidase (GSH-Px) activities and increases malondialdehyde (MDA) and reactive oxygen species’ (ROS) levels [44,86]. In addition, IS induces the expression of apoptosis-related genes (Caspase-3, Caspase-8, Bax, A20, and MT) and autophagy-related pathway key proteins (PTEN-induced putative kinase 1 and parkin), increases the ratio of light chain 3-II (LC3-II) to LC3-I in the intestine, all of which leads to apoptosis and autophagy in the SI [44,80].

As mentioned above, IS can lead to a release of a large number of inflammatory cytokines and chemokines by peripheral immune cells. Inflammatory cytokines in the SI and over-recruited immune cells mediate SI damage and decrease ADFI, mediating the destruction of SI integrity and apoptosis of epithelial cells [88]. However, there is little research on the mechanism of the changes of intestinal digestive enzymes and related transporters caused by IS. Some believe that the changes of digestive enzymes and transporters are closely related to the release of glucocorticoids [89]. Thus, in general, IS causes microbial flora disorders, reduces intestinal digestive enzymes and transporters, and damages SI structural integrity, culminating in digestive and absorption disorders. 

### 3.3. Effect of IS on Nutrient Metabolism in Livestock 

Although the reduction of feed intake, digestion, and absorption are the most important factors in IS-induced growth suppression of livestock, the redistribution of nutrients also plays a role [90]. After the regulation of multiple hormones (growth hormone (GH), thyroid hormone, IGF-1, and GC) and cytokines, the body will redirect nutrients originally used for muscle synthesis and growth to the immune system to maintain a highly activated immune system to prevent diseases, resulting in an increase in FCR [90,91,92,93,94,95].

The activation of the HPA axis caused by IS leads to an increase in the secretion of catabolic hormones such as ACTH and CORT and a decrease in the secretion of anabolic hormones such as GH and IGF-1. In addition, a large number of inflammatory cytokines are secreted, which inhibits the synthesis of energy and promotes the decomposition of energy [96]. Mechanistically, the proinflammatory cytokines released by the body during IS directly act on the thermoregulatory center via the blood–brain barrier, resulting in an increase in body heat production and a reduction in body heat dissipation, which results in an increase in body temperature [7,97]. IS increases energy consumption because in thermostatic animals, the metabolic rate needs to be raised by 10–12.5% to increase the body temperature by 1 °C [98].

In carbohydrate metabolism, IS promotes gluconeogenesis and glycogen hydrolysis in the liver, resulting in an increase in glucose production, a reduction in the uptake of glucose in peripheral tissues such as skeletal muscle and myocardium, resulting in an increase in blood glucose concentration. Because a lot of aerobic energy and hence oxygen is required for the immune response, an anaerobic situation is created that results in the conversion of glucose into lactic acid [3,52].

In protein metabolism, IS increases skeletal muscle protein degradation rate, reduces skeletal muscle protein deposition, speeds up peripheral protein decomposition, enhances liver degradation of valine, leucine and isoleucine, and produces a large amount of liver acute phase protein (APP). In addition, the induction of inflammatory cytokines upregulates the expression of liver proteins involved in immune defense function, amino acid catabolism, ion transport, wound healing, and hormone secretion [92,99].

In lipid metabolism, IS induces the degradation of a large number of lipoproteins in the body and produces a series of inflammatory factors such as TNF-α, which can inhibit the synthesis of fatty acids (FA) in adipose tissue and promote fat degradation [100]. TNF-α also stimulates the liver to synthesize FA de novo and lipolysis in adipose tissue. In addition, the AMP-dependent protein kinase (AMPK) lipid metabolism pathway is activated, inhibiting the activity of acetyl-CoA carboxylase (ACC), thereby reducing the conversion of malonate-CoA. This activates carnitine palmitoyltransferase-1 (CPT-1), which in turn enhances the activity of CPT-1 and the expression of peroxisome proliferator-activated receptor-α (PPAR-α) mRNA, thereby stimulating liver lipid metabolism, accelerating FA oxidation and reducing fat deposition [101].

Thus, IS reduces the appetite of livestock, redistributing nutrients in the body to the immune system, resulting in a decline of ADFI, enhancing catabolism, reducing intestinal digestion and absorption function, all of which finally leads to a decline in growth performance of livestock and poultry, resulting in economic loss to the livestock industry.

### 3.4. Other Effects of IS on Livestock 

IS can cause the activation of the OPG/RANKL pathway in tibia and increase the expression of inflammatory cytokines. Furthermore, IS can enhance the generation of osteoclasts, leading to impairment of bone development [102]. IS can reduce chicken muscle pH, which can result in pale muscle [95]. Mechanistically, IS can lead to accumulation of lactic acid in muscle, resulting in a lowering of muscle pH, which can lead to an increase in protein denaturation and muscle fiber damage, damaging the hydration and texture of meat [103]. In addition, protein denaturation may lead to a decrease in light transmittance to the meat’s surface, resulting in pallor meat [104]. IS can also lead to a sharp decline in lactation of cows [105] due to inhibition of the release of prolactin, GH, and IGF-I, resulting in reduced milk quality and yield [106]. LPS-induced IS decreases the feed intake and egg laying rate in hens, leading to a significant increase in eggshell thickness, strength, and a reduction in albumin quality and content [107].

## 4. Prevention and Control Technologies for IS

Because of intensive farming, the frequency of IS has increased, resulting in a negative impact on the animal industry. It is particularly important to develop safe and effective techniques to prevent and alleviate the effects of IS. Controlling stress, improving management, and resisting the stress response by nutritional intervention are the most popular methods used currently.

### 4.1. Vaccination Program

In order to determine which kind of vaccines are necessary for an individual farm, it is important evaluate the endemic infectious agents in the region, the age of the animal, genetic and health status of the breeding animals, the distance to other farms, and the level of biosecurity to be implemented in the farm [108]. In poultry production, vaccines for Newcastle disease (ND), infectious bronchitis, and infectious bursal disease are used in most countries [109]. Furthermore, vaccine programs should be designed based on other impact factors such as interaction between different vaccines, interference with maternal antibodies, vaccine type, vaccination method, and vaccination frequency. Unnecessary vaccination should be avoided to prevent excessive stimulation of the animal immune system [109,110].

Combined immunization is a scheme worthy of consideration. Combined immunization refers to simultaneous injection with two or more antigens in combination to reduce animal stress [111]. Some have used an inactivated vaccine against serotype 4 and 8 bluetongue disease to immunize sheep and found that simultaneous double-injected booster vaccination yields the highest median serotype-specific neutralization antibody 26 weeks after the first vaccination and a positive serum antibody level at a maximum even after one year [112]. Trivalent vaccine (*Mycoplasma hyopneumoniae*, porcine circovirus type 2 (PCV2) and porcine reproductive and respiratory syndrome virus (PRRSV)) used in pigs has a protective effect on the infection of three pathogens, and the effect is similar to that of a monovalent vaccine, and their growth performance is superior to the uninfected pigs [113]. Similarly, a combined vaccine (*Mycoplasma hyopneumoniae* and PRRSV) is effective in pigs [114]. A recombinant herpesvirus of turkey laryngotracheitis vaccine (rHVT) combined with chicken embryo origin laryngotracheitis vaccine (CEO) provides stronger protection than rHVT alone does. rHVT can reduce the virulence return and replication of CEO, which is safer than CEO alone. rHVT–CEO vaccination strategy is another way to achieve better disease control [115]. 

In general, combined immunization can exert a positive effect without interfering with the effects of each vaccine and reduce the number of immunizations, reducing cross-infection, stress, and improving health and production efficiency.

### 4.2. Improved Feeding Regime

Improving the hygiene of the feeding environment and strengthening farm management can minimize the occurrence of IS. Good hygiene conditions can reduce the contact chance between the animal and pathogenic microorganisms in the environment, thereby reducing the excessive stimulation of the host immune system [116]. A decrease in feeding density has also been used to prevent IS. High-density feeding has been shown to increase IS and thereby affect the growth performance of geese. A reasonable stocking density is not only good for animal welfare but also for better animal growth [117]. Currently, other measures such as “all-in, all-out”, group feeding, and improvement in uniformity of animals are widely implemented in poultry farms, which have reduced the chance of poultry contact with bacteria and viruses and thereby reduced the occurrence of IS. 

### 4.3. Nutritional Regulation

Alleviating animal IS through nutritional regulation has been highlighted in recent years because animals consume more when under IS. This is because of the amount of nutrients required to synthesize immune effector molecules [118]. The addition of amino acids, fats, vitamins, and plant extracts in feed have been shown to alleviate IS and enhance host resistance to IS.

#### 4.3.1. Amino Acid Additives

IS not only increases amino acid requirements of the host but also affects amino acid balance. Adding L-arginine to the diet can effectively alleviate the damage to the intestinal mucosal barrier caused by IS in order to maintain intestinal integrity [54,119]. Mechanistically, L-arginine inhibits the TLR4 signaling pathway, decreases the percentage of CD14^+^ cells with resultant overexpression of proinflammatory factors in animals in IS [9]. In addition, other amino acids and their derivatives, such as glutamic acid [76], glutamate precursor α-ketoglutarate [120,121], L-theanine [122], cysteine [123], N-acetylcysteine [77,124,125], glycine [126], glycyl-glutamine [88], glutamine [127], and asparagine [128,129] have been used as anti-IS additives in feed.

#### 4.3.2. Fatty Acid Additives

Polyunsaturated FA can relieve IS by reducing the release of inflammatory factors. Adding conjugated oleic acid to the diet can inhibit the production of IL-1β, enhance proliferation of lymphocytes by inhibiting the expression of PGE2, and thereby improve the growth performance of pigs [130]. n3 FA can also reduce the expression of PGE2 and the level of inflammatory cytokines in the serum induced by LPS [131]. Fish oil can reduce the release of inflammatory cytokines and improve the growth performance in IS pigs [47,132]. While adding fish oil to the diet can increase cellular immunity and reduce the inflammation index in animals exposed to IS [133], increasing the ratio of n3 to n6 fatty acids in the diet can inhibit the inflammatory response, thereby improving the animals’ performance [134].

#### 4.3.3. Vitamin Additives

Vitamins are necessary for animal metabolism and growth, acting in a variety of ways. Vitamins act to minimize stress, inflammation, and, in general, immune function regulation. Vitamin C can regulate the inflammatory response and oxidative stress caused by LPS and reduce hippocampal cell apoptosis [135]. Vitamin C can also alleviate the damage caused by IS in piglets [136]. Under stress conditions, the requirements for vitamin C are increased, and therefore it is widely used in animal feed to alleviate the health impacts of IS. Adding high doses of vitamin E can increase the feed intake of laying hens during IS and also increase the level of antibodies after vaccination for NDV and avian influenza virus [137]. Vitamin E can inhibit the increase in proinflammatory cytokines, PGE2, and cortisol caused by IS [131,138]. In vitro studies have shown that vitamin A can improve intestinal barrier integrity and reverse LPS-induced intestinal barrier damage by enhancing the expression of tight junction proteins [139].

#### 4.3.4. Trace Element Additives

Trace elements affect immune regulation and can function to reduce stress. For example, MgSO_4_ significantly attenuates LPS-induced acute lung injury, apoptosis, oxidative stress (reducing MDA levels), and lung inflammation [140] and inhibits the NF-κB signaling pathway in mice [141]. Chromium yeast can effectively reduce IL-1β, TNF-α, and COR in piglet serum, thereby alleviating IS [142]. Clinoptilolite (aluminosilicate containing sodium, potassium, and calcium) can inhibit the infiltration and hyperactivation of neutrophils in broilers and reduce plasma and SI mucosal inflammatory cytokines, thereby reducing the inflammatory response caused by LPS [10]. Copper also has an anti-IS additive effect [143,144].

#### 4.3.5. Probiotics Additives

Probiotics, also known as microecological preparations, are a group of active microorganisms or metabolites that benefit the host. The role of probiotics is to improve intestinal health and immune function by regulating the colonization and composition of the intestinal flora, intestinal pH, and also improving communication through the intestinal–brain axis pathway [145]. Feeding some probiotics (*Lactobacillus acidophilus*, *Lactobacillus casei*, *Bifidobacterium thermophilum*, and *Enterococcus faecium*) in the diet can alleviate intestinal inflammation, oxidative stress, and morphological damage caused by enterotoxigenic *E. coli* and enhance the growth performance of piglets [146]. Adding *Bacillus subtilis* to broiler diets can normalize the expression of gut barrier-related genes (JAM2, occludin, ZO1, and MUC2) under IS [147]. Fermentation products of *Bacillus subtilis* not only reduce the expression of LPS-induced SI inflammation genes but also enhance the expression of intestinal barrier genes, thereby improving the growth performance of broilers under IS [58]. In addition, probiotics such as *Bacillus amyloliquefaciens* [55], *Lactobacillus casei* [148], *Lactobacillus acidophilus* [149], *Lact**obacillus delbrueckii* [150], *Enterococcus faecalis* [151], *Clostridium butyricum* [151], and also their fermentation products, can alleviate IS. 

#### 4.3.6. Plant Extract Additives

Several low-molecular-weight secondary metabolites of plants can also play a role in modulating immune stress along with their antibacterial, antiviral, anti-inflammatory, and anti-tumor functions. Since many plant extracts exhibit low toxicity, high safety, and health improvement functions, it has become a research hotspot in the field of feed additives.

Table 1 shows the more important IS resistance plant extracts reported in recent years. These plant extracts include esters, phenols, polysaccharides, glycosides, flavonoids, alkaloids, and crude plant extracts [16,40,152,153,154,155,156,157,158]. Anti-IS mechanisms of plant extract can be categorized into three types, namely (1) relieving inflammation and oxidative stress by inhibiting inflammatory response and oxidative pathways, (2) maintaining intestinal balance by regulating intestinal barrier-related proteins and promoting intestinal cell proliferation, and (3) maintaining hormone balance by regulating the endocrine system. 

## 5. Conclusions

Overall, IS significantly impacts the host neuro–endocrine–immune axis and the brain–gut axis, which can result in hormone secretion disorders, cell apoptosis, intestinal flora disorders, intestinal barrier destruction, oxidative stress, and metabolic disorders, all of which can compromise the immune function and growth performance of livestock. Therefore, efficient and feasible control measures, such as providing good quality nutritious feed, reducing excessive stimulation of the immune system by organizing vaccine procedures at the appropriate time, and using selected plant extracts and potential new drugs for control, are required, as we are experiencing an upsurge of new infectious diseases in livestock.

## Figures and Tables

**Figure 1 animals-12-00909-f001:**
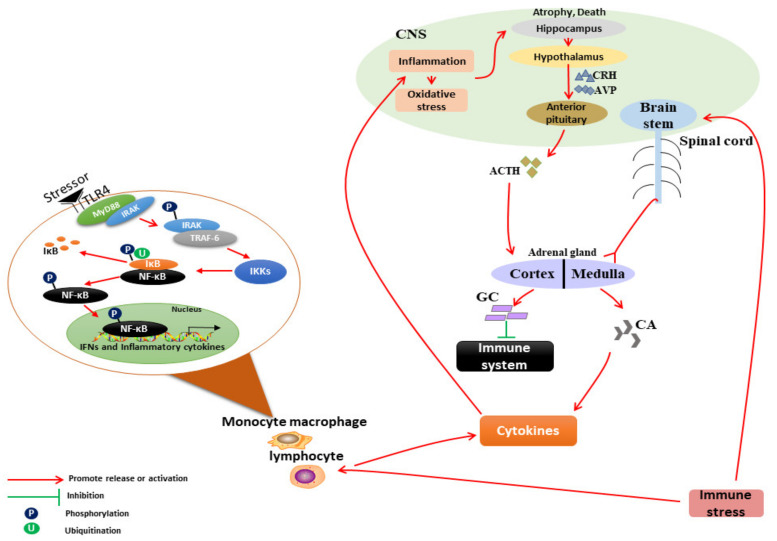
Mechanism of IS on the neuro–endocrine–immune system.

**Figure 2 animals-12-00909-f002:**
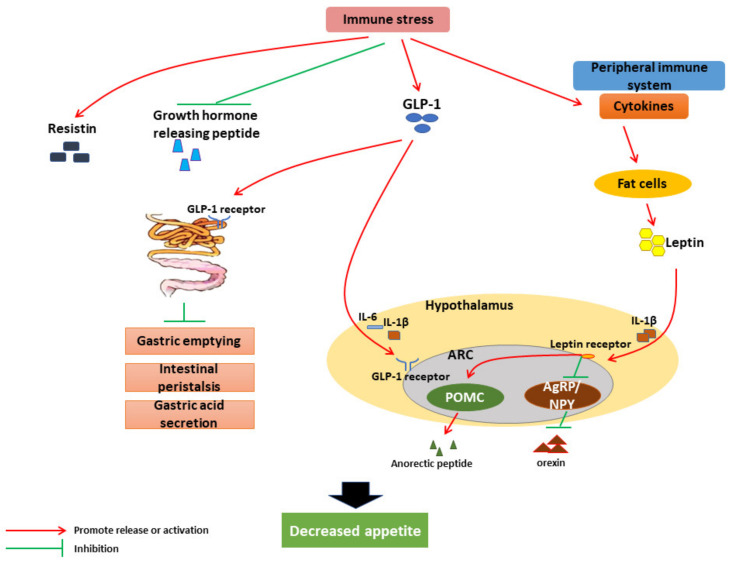
Mechanism of appetite suppression by IS.

**Table 1 animals-12-00909-t001:** A select list of plant extracts that alleviate IS.

Type	Name	Structural Formula	Mechanism of Mitigating IS	Related Literature
Polysaccharides	Acanthopanax senticosus polysaccharide	mixture	Downregulates the HIF-1α/COX-2 pathway and NF-κB to inhibit the release of inflammatory cytokines, increases the activity of diamine oxidase and lactase, improves intestinal morphology, increases GH and IGF-I.	[152,153]
Astragalus polysaccharides	mixture	Inhibits the TLR4/NF-κB signaling pathway, downregulates inflammatory cytokines, and upregulates intestinal tight junction proteins.	[16]
Ginseng polysaccharides	mixture	Inhibits the TLR4/NF-κB signaling pathway, downregulates inflammatory cytokines, upregulates intestinal tight junction proteins.	[16]
Seaweed polysaccharides	mixture	Downregulates IL-1β and IL-6, increases breast milk IgG content, improves intestinal morphology.	[159]
Artemisia ordosica polysaccharide	mixture	Reduces LPS-induced oxidative stress by inhibiting Nrf2/Keap1 and TLR4/NF-κB pathways.	[160]
Glycyrrhiza polysaccharide	mixture	Downregulates the expression of IL-1β and IFN-γ, increases SOD activity, reduces MDA content.	[161]
Glycosides	Stevioside	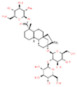	Reduces MDA content and increases antioxidant enzyme activity.	[162]
Salidroside	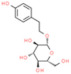	Reduces the levels of NE and 5-HT in the prefrontal cortex, upregulates the BDNF/TrkB signaling pathway, inhibits inflammatory cytokines.	[163]
Hesperidin	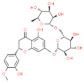	Enhances the activity of monocytes and macrophages, enhances the ratio of intestinal V/C.	[154]
Phenols	Procyanidin	mixture	Inhibits the activities of inflammatory factors (IFN-γ, IL-1β, IL-2, IL-4, IL-6, and IL-10) and nitrogen oxides (NOx).	[164]
Thymol	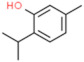	Enhances the barrier function of epithelial cells, reduces the production of ROS and expression of proinflammatory cytokine genes.	[78]
Sesamol	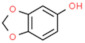	Reduces MCP-1, inhibits NF-κB and inflammatory cytokines TNF-α and IL-1β, prevents lipid peroxidation in serum and liver, increases catalase and glutathione reductase activities.	[165,166]
Curcumin	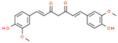	Regulates the JNK/NF- κ B/Akt signaling pathway, plays an anti-inflammatory and antioxidant effect, may alleviate liver damage and liver lipid metabolism disorders by increasing m^6^A RNA methylation.	[155,156]
Green tea polyphenols	mixture	Inhibits NF-κB signaling and inhibit NLRP3 inflammasome activation.	[167]
Carvacrol	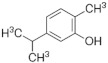	Downregulates the expression of inflammatory cytokines by inhibiting the TLRs/NF-κB pathway.	[168]
Flavonoids	Artemisia argyi flavonoids	mixture	Decreases the expression of NF-κB, IL-1β, and IL-6.	[45]
Quercetin	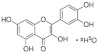	Inhibits the activity of IκBα and NF-κB into the nucleus, reduces the expression of TNF-α and IL-1β.	[157]
Genistein	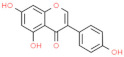	Enhances the activity of monocytes and macrophages, increases the ratio of intestinal V/C.	[154]
Esters	Ellagic Acid	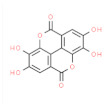	Antioxidant activity, inhibits AChE activity.	[27]
Sesame lignans	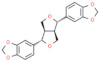	Reduces MCP-1, inflammatory cytokines TNF-α and IL-1β, prevents lipid peroxidation.	[165]
Alkaloids	Berberine	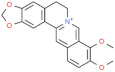	Inhibits NF-κB signal transduction and the expression of inflammatory mediators, enhances the activity of antioxidant enzymes.	[158]
Crude extract	Artemisia argyi aqueous extract	mixture	Inhibits the release of CORT and IL-2.	[40]

## Data Availability

Not applicable.

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
