# Peer review of "Effect of Immune Stress on Growth Performance and Immune Functions of Livestock: Mechanisms and Prevention"

_animals, 2022, doi:10.3390/ani12070909_

Round 1

Reviewer 1 Report

This review seems to be well written in general, however, there are some typos and the description of some parts (Line 113-117) are very poor.

Especially in References, almost all articles were incompletely written. That makes me concern about reliability of this review.

  1. At first all references should be reconstituted properly in order to MDPI style.

An example is as follows, Family name, Initial of own name. Title of article. Abbreviated journal name in italic, Years. Volume, Pages.

  1. Totally, these citations seem to be biased. You should delete unnecessary citations.
  2. There were so many inadequate citations. Some are incomplete (no volume, no pages), so that very difficult to reach original article.
  3. Some articles seem to be not necessary and should be deleted. When you think that are important you better to emphasize them in the review, otherwise delete.

These concerns must be properly addressed by the authors. If without such improvement, I don’t agree this MS for publication.

Author Response

We are thankful to the reviewers for providing very valued input. Based on these comments, the manuscript has been revised further. We believe the inclusion of new suggestions has substantially improved the quality of this manuscript. Below we provide our point-by-point response to each comment and have included these changes in the manuscript.

Response to reviewers comments

Response to Reviewer #1 comments:

This review seems to be well written in general, however, there are some typos and the description of some parts (Line 113-117) are very poor.

Especially in References, almost all articles were incompletely written. That makes me concern about reliability of this review.

Q1. There are some typos and the description of some parts (Line 113-117) are very poor.

Thank you. We have modified it to “The effect of IS on animal immune function is bidirectional. It may enhanced or suppresses immune function, which depends largely on stress intensity and duration of stress[37]. It is generally believed that short-term (acute) stress can enhance the immune function of livestock while long-term, low intensity (chronic) stress can cause immunosuppression[37,38].”

Q2. At first all references should be reconstituted properly in order to MDPI style.

Thank you. I'm sorry for our mistakes in the references. We have used EndnoteX9 to correct references.

Q3. Totally, these citations seem to be biased. You should delete unnecessary citations.

Thank you. We have deleted unnecessary citations.

Q4. There were so many inadequate citations. Some are incomplete (no volume, no pages), so that very difficult to reach original article.

Thank you. We have modified it, but one of the papers (doctoral dissertation) is from CNKI. Here is a link to its original article.

https://kns.cnki.net/kcms/detail/detail.aspx?dbcode=CDFD&dbname=CDFDLAST2016&filename=1015732683.nh&uniplatform=NZKPT&v=olwDmTUQU_BcCzFyrvYrcRsi_yDELhRJqPZZ8C-XStzPiYrhReUGjQEXUONKbNnD

Q5. Some articles seem to be not necessary and should be deleted. When you think that are important you better to emphasize them in the review, otherwise delete.

Thank you. We have modified it.

Reviewer 2 Report

I have not found significant limitations in this proposal; conversely, I think that it has many strengths, such as originality and methodological accuracy. I think that this project I reviewed reports novel findings that will be useful for poultry industry, health and well being. Congratulations, great review.

Please find additional comments in the attachment.

Author Response

Thank you for your affirmation of the article, and we will continue to work hard.

Round 2

Reviewer 1 Report

I agree to accept this revised form of MS at now.